# Canola with Stacked Genes Shows Moderate Resistance and Resilience against a Field Population of *Plasmodiophora brassicae* (Clubroot) Pathotype X

**DOI:** 10.3390/plants12040726

**Published:** 2023-02-06

**Authors:** Nazmoon Naher Tonu, Rui Wen, Tao Song, Xiaowei Guo, Lee Anne Murphy, Bruce Dean Gossen, Fengqun Yu, Gary Peng

**Affiliations:** 1Saskatoon Research and Development Centre, Agriculture and Agri-Food Canada, 107 Science Place, Saskatoon, SK S7N 0X2, Canada; 2Pest Surveillance Initiative, 5A-1325 Markham Road, Winnipeg, MB R3T 4J6, Canada

**Keywords:** canola, rapeseed, clubroot, gene pyramiding, pathotype, resting spores, resistance durability

## Abstract

Genetic resistance is a cornerstone for managing clubroot (*Plasmodiophora brassicae*). However, when used repeatedly, a clubroot resistance (CR) gene can be broken rapidly. In this study, canola inbred/hybrid lines carrying one or two CR genes (*Rcr1/CRa^M^* and *Crr1^rutb^*) were assessed against *P. brassicae* pathotype X by repeated exposure to the same inoculum source under a controlled environment. Lines carrying two CR genes, either *Rcr1* + *Crr1^rutb^* or *CRa^M^* + *Crr1^rutb^*, showed partial resistance. Selected lines were inoculated with a field pathotype X population (L-G3) at 5 × 10^6^ resting spores/g soil, and all clubs were returned to the soil they came from six weeks after inoculation. The planting was repeated for five cycles, with diseased roots being returned to the soil after each cycle. The soil inoculum was quantified using qPCR before each planting cycle. All lines with a single CR gene were consistently susceptible, maintaining high soil inoculum levels over time. The lines carrying two CR genes showed much lower clubroot severity, resulting in a 10-fold decline in soil inoculum. These results showed that the CR-gene stacking provided moderate resistance against *P*. *brassicae* pathotype X, which may also help reduce the pathogen inoculum buildup in soil.

## 1. Introduction

Clubroot, caused by the protist pathogen *Plasmodiophora brassicae* Woronin, is an important disease in many brassica crops worldwide and a threat to canola (*Brassica napus* L.) production in Canada; severely damaged canola fields have shown 100% yield losses [1]. Clubroot infection normally starts in root hairs through zoospores released from germinating resting spores, and secondary zoospores released from infected root hairs can penetrate root epidermis, leading to the colonization of cortical tissues by *P*. *brassicae* plasmodia [2]. The infection stimulates hypertrophy and hyperplasia of root tissues, resulting in characteristic clubbing symptoms. As the root matures, the plasmodia are converted into millions of resting spores and are released into the soil, once the clubs (galls) decompose, for a new disease cycle. Some resting spores may survive in soil for many years, but most of them seem to die rapidly [3,4]. Relative to susceptible canola, resistant cultivars with lower disease severity (fewer and smaller galls) tend to return fewer numbers of resting spores into the soil [5,6,7].

Genetic resistance is a cornerstone for clubroot management in canola, as other measures alone are often either insufficient or impractical. Repeated uses of a single CR gene, however, may result in rapid erosion of resistance. In Canada, the first clubroot-resistant canola cultivar (‘45H29′) was introduced in 2009, and carries one of the CR genes from the European winter rapeseed cv. Mendel [8]. More resistant cultivars were registered in following years, which, together with 45H29, were considered ‘first-generation’ clubroot-resistant canola cultivars carrying only a single CR gene [8]. At that time, five pathotypes were identified in the Canadian *P. brassicae* population based on the Williams’ system [9], i.e., pathotypes 2, 3, 5, 6 and 8, with pathotype 3 being dominant in canola [10]. The first-generation cultivars were resistant to all the five pathotypes [11]. In 2013, however, severe clubroot was found on ‘resistant cultivars’ in central Alberta [12], caused by Williams’ pathotype 5, which appeared different from the earlier strains, as they were apparently virulent to 45H29 and other resistant cultivars. These new strains were, therefore, considered ‘novel’ and referred to as pathotype ‘5X’ initially to distinguish them from the original pathotype 5 strains. They eventually were classified as pathotype ‘X’ based on the Canadian Clubroot Differential (CCD) system [13], which also revealed that the *P. brassicae* population was much more diverse in Canada than what was believed before. It has also been shown that these resistant cultivars could be defeated within as few as two cycles of exposure to a single field inoculum source of *P. brassicae* under controlled conditions [14]. The diversity in the pathogen population presents a challenge to resistance durability.

In general, there appeared only limited CR sources, mostly from European turnips [15]. Stacking resistance genes is a common way to broaden disease resistance against multiple pathogen races/pathotypes, potentially extending the useful life of individual resistance genes, as shown in rice against bacterial blight caused by the bacterial pathogen *Xanthomonas oryzae* pv. *oryzae* [16]. In canola, however, the effect of pyramid major *R* genes for long-term management of blackleg (*Leptosphaeria maculans*) appeared complicated; it may depend on the molecular mechanisms by which the pathogen overcomes the resistance [17], and combining specific *R* genes with quantitative resistance would render canola cultivars more durable in their resistance to the disease compared to using the major *R* genes alone [18]. For clubroot, pyramiding three CR genes in Chinese cabbage (*B*. *rapa*) expanded the resistance against multiple pathotypes compared to cultivars carrying only a single CR gene [19]. However, there has been no reported study on resistance durability associated with CR-gene stacking.

The CR gene derived from Mendel (on chromosomes A03) appeared to be present in most of the first-generation resistant canola cultivars and was effective against the initial group of *P. brassicae* pathotypes found in Canada [8,13]. Subsequently, CR genes on chromosome A08 were found resistant to some of the novel pathotypes [20,21], although efficacy varied depending on the pathotype and field collection of *P*. *brassicae*. The A08 CR gene(s) have recently been pyramided with those on A03 in several Canadian canola hybrids (‘second generation’) for a broader range of efficacy. It was unclear, however, whether the CR-gene stacking would also provide a durable form of resistance vs. a single CR gene alone, especially against some of the newly identified pathotypes virulent towards the first-generation resistant cultivars. The objective of this study was to evaluate clubroot responses of canola inbred/hybrid lines that carried single and multiple CR genes to field collections of the novel pathotype X, as well as the durability of moderately resistant lines based on two CR genes (A03, A08) against a field collection (L-G3) of pathotype X over multiple cycles of exposure under controlled-environment conditions as a case study. The information may help understand the potential of CR-gene stacking for sustainable clubroot management.

## 2. Materials and Methods

### 2.1. Resistance to Pathotype X

Prior to the study, a total of 20 canola (*B. napus*) varieties/lines (Table 1) were developed at Nutrien Ag Solutions, Saskatoon, Saskatchewan, Canada, using doubled haploid (DH) lines carrying one or more CR genes from three sources. One was a CR gene from rutabaga that had been mapped to the same region on chromosome A08 as CR genes *Crr1* and *Rcr3* [20,21,22,23]. This region was considered to contain two CR genes based on the fine mapping of a 1.6 cM genetic region [22], but it was not clear whether these CR genes mapped to this region were allelic. To avoid nomenclature confusion, this rutabaga-derived CR gene is referred to as *Crr1^rutb^* for this study. The other two resistance sources showed CR genes mapped to a close range on chromosome A03: *Rcr1* from *B. rapa* ssp. *chinensis* [24] and a CR gene derived from cv. Mendel developed from a resynthesized *B*. *napus* line made with a cross of the European Clubroot Differentials (ECD) ECD04 (*B. rapa*) and ECD15 (*B*. *oleracea*) [25]. Fine mapping by Fredua-Agyeman et al. [8] showed that 12 markers from the genomic region of A03 that houses the CR locus *CRa/CRb^Kato^* [26,27,28,29] were closely associated with the resistance derived from Mendel, with <2% recombination rates. Therefore, this Mendel-derived CR gene is referred to as *CRa^M^* for this study.

Initially, the CR genes *Rcr1*, *CRa^M^* and *Crr1^rutb^* were introgressed respectively into spring canola breeding lines using recurrent backcrosses and marker-assisted selection, which were later made into DH lines at Nutrien Ag Solutions, with a total of 20 inbred/hybrid lines produced (Table 1) carrying different CR genes or CR-gene combinations also using markers for *Rcr1*, *CRa^M^* and *Crr1^rutb^* [8,20,24]. Two of the lines (14 and 15) were produced by crossing a female DH line carrying the CR gene(s) in A08 with the male line carrying a CR gene in A03 (*Rcr1* or *CRa^M^*). Other lines, carrying either a single or multiple CR genes, were produced by crosses where the line carrying the A08 CR gene(s) was used as a male donor. Each of these lines showed consistent resistance to *P. brassicae* pathotype 3, the predominant pathotype in western Canada, as well as to several old (based on the Williams’) and new (on Canadian Clubroot Differentials [13], CCD) pathotypes found in Canada (data not shown), but the evaluation on resistance to a broad range of CCD-characterized pathotypes is still ongoing.

Three collections of pathotype X (L-G1, L-G2 and L-G3) of *P. brassicae* from fields in Alberta, Canada were selected for the current study. These collections were made from commercial fields where first-generation resistant canola cultivars developed severe clubroot [13]. Each collection was increased separately in the canola cultivar 45H29, which was resistant to initial pathotypes 2, 3, 5, 6 and 8 (Williams’) identified in Canada, but susceptible to pathotype X [13].

To select canola lines for the assessment of resistance durability, these 20 lines were evaluated initially for responses to the three collections of *P. brassicae* pathotype X under controlled-environment conditions following the protocols of Chu et al. [24]. Briefly, clubs stored at −20 °C were homogenized in water using a commercial blender, and the suspension filtered through eight layers of cheesecloth. The concentration of resting spore suspension was estimated using a haemocytometer and adjusted to 5 × 10^7^ spores/mL. The Sunshine #3 potting mix (Sun Gro Horticulture, Vancouver, BC, Canada) was amended with Osmocote plus a 16-9-12 (N-P-K) fertilizer (Scotts Canada, Mississauga, ON, Canada) applied at about 12.5 g/L, and the amended potting mix had a final pH of 6.3. The amended potting mix was placed in tall, thin plastic cones (20-cm tall, 5-cm diameter; ‘conetainers’, Steuwe & Sons, Corvalis, OR, USA) and seeded with two canola seeds per pot. The potting mix in each seeded pot was packed and watered thoroughly, then 5-mL resting-spore suspension was added to produce about 1 × 10^6^ spores/g mix. Each pot was thinned to one plant at the cotyledon stage. The susceptible cultivar ‘Westar’ and the ‘first-generation’ resistant cultivar 45H29 were included as controls. The use of this amended potting mix, as well as the inoculation protocol, allowed consistent clubroot symptom expression at four weeks post seeding.

A completely randomized design was used, with seven plants (replicates) per canola line. Due to the space limitation in the growth room and scheduling challenge, only the test against L-G3 was repeated. For the test against L-G2, a total of 18 replicates were used for each canola line when additional space became available unexpectedly. Inoculated cones were maintained in a growth room set at 22/16 °C (day/night) with a 16 h photoperiod (280–575 μmol/m^2^s^1^) for clubroot infection and development. The potting mix was kept saturated for the first week following inoculation, and plants were watered regularly afterwards to keep the mix moist. These conditions, including the ambient temperatures, soil pH and moisture are highly conducive to clubroot infection and symptom development, and have been used continuously in our lab for over 15 years. Five weeks after inoculation, plants were removed from their pots and the roots were washed and assessed for clubroot severity using a 0–3 scale [30], where 0 = no clubroot symptom, 1 = mild clubbing, mostly on branch roots, 2 = moderate clubs and 3 = large clubs formed on the main root, as on Westar. A pictorial key for the 0–3 scale has been shown previously [31]. A disease severity index (DSI) could be calculated using the formula below:(1)$$DSI=\frac{\sum \left[\left(rating\;class\right)\times \left(\mathrm{number}\;\mathrm{of}\;\mathrm{plants}\;\mathrm{in}\;\mathrm{the}\;\mathrm{class}\right)\right]}{\left(\mathrm{total}\;\mathrm{number}\;\mathrm{of}\;\mathrm{plants}\right)\times 3}\times 100\%$$

### 2.2. Durability of Resistance

Resistance durability was examined for moderately resistant lines 14 and 15 against the field collection L-G3 of pathotype X; these two lines carry CR genes from both A03 (*Rcr1* or *CRa^M^*) and A08 (*Crr1^rutb^*) chromosomes, and was assessed by repeated exposure to the same inoculum source of L-G3 in comparison with five susceptible lines with a range of CR-gene compositions (Table 2). The experimental protocol differed in several ways from those above; first, the cones holding individual canola plants were replaced with large pots (20-cm diameter) with multiple plants. Second, the large pots were filled with infested potting mix by mixing a resting-spore suspension with the potting mix in a tub before potting and seeding. Third, a higher level of *P*. *brassicae* inoculum (~5 × 10^6^ spores/g) was used for the first cycle of exposure to ensure a level of infection on the moderately resistant lines 14 and 15.

Each large pot was filled with the infested potting mix and planted with 23 seeds per canola line to form an experimental unit. A saucer was placed under each pot to collect excess water and keep the soil moist. Two Westar (highly susceptible) seeds were also planted in each pot, with locations marked, to verify that conditions were conducive for infection. The study was arranged in a randomized complete block design with three replicates (pots). Watering, fertilization and disease assessment were the same as described previously.

At six weeks after seeding, plants were assessed individually for clubroot severity using the 0–3 scale. Westar plants generally developed large galls on the main root just below the soil surface; this indicated that conditions in the study were suitable for infection. Galls were rated relative to those on Westar and DSI calculated for each pot of plants. After clubroot assessment, the roots and galls of Westar plants were discarded and all roots of the test lines (clubbed or not) were dried lightly at room temperature for 24 h, then buried in the same pot of soil they came from for 3 weeks to allow resting spores to mature and galls to decompose. The soil in each pot received 200 mL of water initially and again 10 d later to keep the soil damp until galls became delicate and easy to break up. Then the soil in each pot was thoroughly mixed to redistribute the resting spores uniformly.

The initial inoculation using ~5 × 10^6^ spores/g inoculum was the first cycle of exposure. For the subsequent four cycles of exposure, the process was repeated; each pot was seeded with the same canola line as in the previous cycle (including Westar) and the plants were maintained for six weeks. After disease rating, the roots of the test lines were dried slightly and buried in damp soil for three weeks to allow the maturation of resting spores. Resting spores from decomposed galls were mixed thoroughly into the soil. Before seeding a new planting cycle, about 10 g of potting mix was collected at five random locations from each pot (replicate), and stored at −20 °C until being used for resting-spore quantification. Shortly after seeding the 3rd cycle in the repeated experiment run, canola plants became heavily infested by aphids, so greenhouse staff applied the insecticides Kontos (spirotetramat) and Intercept (imidacloprid) as a soil drench following the product labels. Both insecticides are used commonly for the control of a range of insect pests in greenhouses and nurseries, and have been a standard treatment in our greenhouse for years for the control of aphids. However, their impact on the clubroot infection was unknown. The experimental process is illustrated in Figure 1 below. The experiment was repeated once, with a total of six replicates.

### 2.3. Quantification of Resting Spores

The concentration of resting spores in soil samples taken at the beginning of each planting cycle was quantified using quantitative PCR (qPCR) for five of the canola hybrid/inbred lines with variable resistance to pathotype X, i.e., the moderately resistant lines 14 (*Rcr1* + *Crr1^rutb^*) and 15 (*CRa^M^* + *Crr1^rutb^*), and the susceptible lines 6, 16 and 20 carrying a single CR gene *Rcr1*, *CRa^M^* and *Crr1^rutb^*, respectively. The qPCR protocol was similar to what was described by Rennie et al. [32], which has also been used previously for quantification of *P*. *brassicae* resting spores in field soils [3,4,33]. Briefly, samples were air dried and a 0.25 g subsample was ground to a fine powder with a mortar and pestle that were cleaned after each use. The total DNA in these soil samples was extracted using PowerSoil^®^ DNA Isolation Kits (Qiagen, Toronto, ON, Canada), as per manufacturer’s instructions, and DNA samples were serially diluted with sterile deionized water (sdH_2_O) to minimize the interference by PCR inhibitors in the soil and kept at −30 °C until use. The primers DC1F (5′-CCTAGCGCTGCATCCCATAT-3′) and DC1R (5′-CGGCTAGGATGGTTCGAAAA SYBR-3′) were used for PCR reactions on a ViiA7 RT-PCR system (Applied Biosystems, Mississauga, ON, Canada) in a 20-µL reaction volume. A melting point analysis was conducted at the end of each reaction, a single amplification product was confirmed, and the number of *P. brassicae* resting spores calculated using a standard curve [32,34].

For qPCR quantification, these soil samples were selected randomly for testing in batches with three technical replicates per sample. The sensitivity of the qPCR protocol was assessed by subjecting eight randomly chosen soil samples to cell lysis on the FastPrep^®^-24 homogenizer (MP Biomedicals, Solon, OH, Canada) prior to DNA extraction, and the inoculum amounts (based on the DNA extracted) were quantified using qPCR and droplet-digital PCR (ddPCR, Bio-Rad Canada, Mississauga, ON, Canada). A previous study showed that the cell lysis and use of ddPCR could improve the sensitivity of resting-spore quantification in soil samples [34].

### 2.4. Data Analysis

All statistical analysis was performed using the SAS version 9.3 (SAS Institute, Cary, NC, USA). DSI data for L-G3 inoculations were transformed (arcsine square root) prior to statistical analysis for compliance of normal data distribution based on Shapiro–Wilk Test (PROC UNIVARIATE), and back-transformed for presentation. The homogeneity of variance was checked with Levene’s Test prior to pooling data from different repetitions. As the DSI data from the study of resistance durability were not homogeneous, they were analyzed separately for ANOVA. Fisher’s Protected LSD was used to separate the means only when ANOVA was significant (*p* ≤ 0.05).

qPCR estimates of resting spores in soil samples were log_10_ transformed prior to analysis, and the transformed data showed a normal distribution based on the Shapiro–Wilk Test. Estimated resting-spore concentrations were analyzed also using ANOVA and protected LSD, followed by regression analysis of each canola line over the five planting cycles.

## 3. Results

### 3.1. Resistance to Pathotype X

All three collections of pathotype X (L-G1, L-G2 and L-G3) produced severe clubroot on the susceptible control cultivars Westar, as well as on 17 of the 20 inbred/hybrid lines assessed with the clubroot severity >50% of that on Westar (Figure 2 and Appendix A). Of the remaining three lines, lines 14 and 15 were resistant (severity <30%) or partially resistant (severity >30% but <50%) when compared to Westar, while line 20 was resistant or partially resistant to L-G1 and L-G2 (Appendix A) but susceptible to L-G3 (Figure 2).

Several lines that carried the same CR genes differed in resistance to pathotype X, depending on how the crosses were made and on the specific pathotype collection used for inoculation. For example, line 15, which was produced in a cross with *Crr1^rutb^* from the female parent and *CRa^M^* from the male parent, was partially resistant to all three pathotype X collections. In contrast, lines 1 to 8, which were the progeny of reciprocal crosses (*Crr1^rutb^* from the male parent, *CRa^M^* from the female parent), were consistently susceptible (Figure 2 and Appendix A).

### 3.2. Durability of Clubroot Resistance

Seven selected inbred/hybrid canola lines were continuously exposed to the L-G3 over the course of five planting cycles in a controlled-environment study. All roots from the previous cycle (clubbed or not) were returned to the soil of the same experimental unit (pot) to add the inoculum for the next cycle. In the initial run of the study, DSI was generally above 60% for the susceptible controls (lines 6, 12, 13, 16 and 20) and the clubroot severity changed little over the five cycles of exposure (Figure 3A). The severity was lower on lines 14 and 15 (partially resistant) relative to the control lines in each of the planting cycles, and DSI decreased (*p* < 0.05) for both lines in cycles 4 and 5 compared to the earlier cycles.

In the repetition of the study, the clubroot severity was higher on all lines in the first cycle of exposure (Figure 3B) compared to the initial run of the study (Figure 3A). DSI declined (*p* < 0.05) substantially in the second cycle on lines 14 and 15, but not on any of the control lines. Clubroot severity was unexpectedly low in the third planting cycle on all lines; many plants showed only minor clubbing symptoms, including Westar plants. The insecticides Kontos and Intercept were applied as a soil drench shortly after the start of this planting cycle for control of heavy aphid infestation; the potential impact of this treatment on clubroot infection and/or symptom development will be discussed in a later section. DSI increased (*p* < 0.05) in cycles 4 and 5 on the control lines but remained low on lines 14 and 15, with DSI ranging only from 2% to 7% (Figure 3B).

### 3.3. Quantification of Spores

qPCR quantification of resting spores in soil were carried out for five of the canola lines prior to each planting cycle to determine changes in soil inoculum over time. Mixing a suspension of resting spores with the potting mix resulted in 8 × 10^6^ spores/g soil in the initial run and 2–3 × 10^6^ spores/g in the repetition of the study, despite the target initial spore concentration of 5 × 10^6^ spores/g soil. The pattern of change in resting spore concentration was similar between the two study runs, and because of the homogeneity of variance, the data were combined for analyses. No significant change was detected in spore concentration for the susceptible controls (lines 6, 16 and 20) over the five planting cycles (Figure 4), but spore numbers declined for lines 14 (*r* = 0.91, *p* < 0.01) and 15 (*r* = 0.64, *p* < 0.05).

Cell lysis using the FastPrep^®^ homogenizer enhanced the amount of DNA extracted from the *P. brassicae* resting spores only slightly, but the pattern of qPCR estimation was similar between samples with and without cell lysis. DSI also declined (*p* < 0.05) on these partially resistant lines in cycles 4 and 5 compared to earlier planting cycles. Overall, repeated planting of these resistant lines resulted in an almost 10-fold decline in resting-spore numbers over the five planting cycles. In contrast, the concentration of resting spores did not change substantially when susceptible lines were grown continuously in the same soil.

Both DSI and spore concentration trended lower for lines 14 and 15 over the five planting cycles in the two repetitions (Figure 4 and Appendix A). However, there was no correlation between the two variables. It is noteworthy that DSI dipped suddenly in cycle 3 of the second run of the study on all lines and then increased on susceptible lines in the cycles 4 and 5. During this period, however, the spore concentration changed little, and remained >10^5^ spores/g in every soil sample.

## 4. Discussion

A range of management practices have been examined to reduce the impact of clubroot, but the use of resistant cultivars in combination with a 2- to 3-year crop rotation interval (Peng et al., 2015) appears the most cost-effective strategy for canola production. Several new CR genes have been identified recently [35], but CR sources are limited, and no single CR gene can be effective against all *P. brassicae* pathotypes identified [21,35]. The diversity of the *P. brassicae* population on the Canadian Prairies [13,36] makes it unlikely for any CR gene to be continuously effective over time, as evidenced by the rapid breakdown of first-generation resistant cultivars [13]. The key objective of this study was to assess the partial resistance resulting from *Rcr1* or *CRa^M^* (A03) and *Crr1^rutb^* (A08) stacking for potential durability when exposed repeatedly to a field population of new pathotype, such as pathotype X.

Pyramiding resistance genes is a common approach to broadening disease resistance and extending the effective life of resistance genes [16,37]. However, stacked CR genes may sometimes provide only partial or even variable resistance, depending on the CR source and pathotype population, as observed in this study. The hybrid lines carrying two CR genes (lines 14 and 15) showed suppression of pathotype X, but the same genes used individually appeared insufficient or variable (Figure 1 and Figure 2). It is possible that the partial resistance of lines 14 and 15 is derived from *Crr1^rutb^*, as inbred lines carrying *Rcr1* or *CRa^M^* alone were susceptible to each field population of pathotype X (Figure 1 and Appendix A). However, it should be noted that line 20, carrying the CR gene *Crr1^rutb^*, was resistant merely to L-G2 (Appendix A); this poses questions about how the *Crr1^rutb^* functions against specific field collections of pathotype X.

Mapping of *Crr1^rutb^* [20] showed that this CR gene is located in the region where *Crr1* [22,23] and *Rcr3* [21] also reside on chromosome A08. *Rcr3* is flanked by two SNP markers that are 231.6 Kb apart; this interval contains 32 genes, three of which (*Bra020951*, *Bra020974* and *Bra020979*) have been associated with disease resistance [21]. Previous studies indicate that *Crr1* consists of two loci; a major locus, *Crr1a,* which encodes a TIR-NB-LRR protein, and a minor locus, *Crr1b* [22,23]. Although it cannot yet be determined whether *Crr1^rutb^* is the same gene or allele of *Crr1* or *Rcr3*, there is possibly more than one component involved in the resistance of *Crr1^rutb^*, and all of them may be required for resistance to all three collections of pathotype X. It is possible that one or more component of *Crr1^rutb^* had been lost during the production of line 20, as this may happen often with CR genes in the *Crr1* region when they are carried in male parental lines during hybridization [38]. This may also explain the susceptibility of lines 1–8, all of which carried *Crr1^rutb^* and *CRa^M^* but were produced from crosses where *Crr1^rutb^* was in a male parent. For the production of lines 14 and 15, *Crr1^rutb^* was carried in a female parent in the crosses, and all essential resistance components might have been retained. There may also be complementary interaction between the two CR genes [35] in response to specific field collections of pathotype X, but this will need further confirmation.

In the study of resistance durability, clubroot severity was consistently lower on the partially resistant lines 14 and 15 carrying two CR genes compared to the controls (Figure 2). There was no indication of resistance erosion over the five cycles of continuous exposure; DSI remained low or even declined in these partially resistant lines. This relatively stable resistance was in contrast to the results of a previous study, where two moderately resistant canola cultivars carrying a single CR gene showed substantially reduced efficacy after only two cycles of exposure to a source of field *P*. *brassicae* inoculum [14].

There is an increasing body of evidence indicating that *P*. *brassicae* field populations often consist of multiple genotypes, including rare types, allowing rapid pathogen adaptation to new CR genes deployed [39]. In the current study, the L-G3 was from a field where a first-generation resistant canola cultivar lost efficacy after only 4 years after use [13]; this may indicate that this virulent pathotype was possibly present in the pathogen population prior to the use of the resistant cultivar. In heavily infested fields or patches where millions of billions of spores may be present, there may be greater chances for multiple pathotypes to exist in the soil, and resistance erosion will depend on the type and proportion of rare virulent genotypes present. The diversity of the L-G3 population was unknown, but it might have been reduced by continuous inoculum increases in the cultivar 45H29 carrying *CRa^M^* [38], which would have exerted strong selective pressure for the pathotype X against many other pathotypes. Conversely, stacking *Rcr1/CRa^M^* (A03) and *CRa^rutb^* (A08) might have suppressed more pathotypes, including pathotype X, which otherwise could be virulent or moderately virulent towards any of the CR genes alone, mitigating rapid increases in virulent inoculum in soil.

Under field conditions, severe clubroot on a single crop can contribute up to10^10^ spores/g soil (Hwang et al., 2013). Resistant cultivars, with fewer and smaller clubroot galls, can decrease the inoculum buildup in soil [5,6,40]. In the current study, where inoculated plants were given only six weeks for clubroot development, galls were smaller (Appendix A) relative to those on mature field plants, likely returning fewer resting spores into the soil. Besides, young galls may contain many immature spores [41], which are less effective for infection and often short-lived in soil [3,4,42]; these may explain why the spore concentration did not increase substantially over time for susceptible controls (Figure 3). Despite all these, the 6-week interval used for the assessment may still be valid for these greenhouse trials, as the soil inoculum levels were compared between resistant and susceptible canola lines while all other factors/conditions were kept similar. The results showed that continuous planting of the resistant lines 14 and 15 resulted in an almost 10-fold decline in soil resting-spore concentration (Figure 3), along with reduced clubroot severity over time. This demonstrates that even a moderate level of resistance may reduce soil inoculum buildup when compared to intensive cropping of susceptible canola cultivars.

A high level of inoculum (~5 × 10^6^ spores/g soil) was used in the initial cycle of exposure to ensure the development of consistent clubbing on moderately resistant lines 14 and 15; this was intended to simulate the impact of the double CR-gene hybrids on pathotype X (or possibly any other virulent pathotypes) inoculum in heavily infested fields. Despite the gradual reduction in soil inoculum by these resistant hybrids, the resting spore concentration remained >10^5^ spores/g in all of the soil samples (Figure 3), which is sufficient to cause severe clubroot on susceptible canola [43]. This initial inoculum level, however, was higher than those found in most of the infested commercial fields on the Canadian Prairies [40,44], indicating the potential for moderate resistance to deter inoculum buildup in those fields. On the flip side, lower soil inoculum levels would likely cause lighter clubroot infection [43], resulting in a smaller number resting spores going back into the soil. These would possibly favor CR performance and durability. As demonstrated by earlier studies, a >2-year break from a canola crop can reduce resting spores in soil by up to 90% [3,4]. It is therefore highly advisable to use clubroot-resistant cultivars with some of the culture-control measures, including extended crop rotations and possibly soil liming, to reduce inoculum and disease pressure for greater CR longevity and more sustainable clubroot management.

Although the resting-spore concentration and clubroot severity showed similar patterns on these canola lines (Figure 2 and Figure 3), there was no correlation between the two parameters, possibly due to the sudden dip in DSI in cycle 3 of the repeated experiment. It is possible that this sudden dip in DSI was caused by the soil drenching with insecticides Kontos and Intercept for aphid control, especially the surfactants in these products, which may suppress clubroot infection as shown by Hildebrand and McRae previously [45]. This notion seems to be supported by the fact that the residual effect in the soil extended further into cycles 4 and 5 of this repetition run (Figure 3B), where DSI increased gradually on all susceptible lines, but was still lower than that in cycles 1 and 2. This may serve as a reminder that an alternative method should probably be considered to control aphids for similar experiments in the future.

## 5. Conclusions

In conclusion, the hybrid canola lines 14 and 15 stacked with *Rcr1* or *CRa^M^* (A03) and *Crr1^rutb^* (A08) genes provided moderate resistance to different field collections of *P*. *brassicae* pathotype X, whereas any of the single genes alone was ineffective or showed variable efficacy (Figure 2 and Appendix A). This resistance was possibly derived from *Crr1^rutb^*, which likely has more than one gene, and all of them may be required for a broad range of resistance against pathotype X. This moderate resistance based on CR-gene stacking also appeared durable; clubroot severity was consistently reduced compared to that on susceptible controls over five cycles of exposure to the same source of L-G3 (Figure 3), and there was a 10-fold decline in the soil spore concentration over time (Figure 4). These results demonstrate that CR-gene stacking, even with only moderate levels of resistance, can not only extend the efficacy against pathotype X, but also show the trait of resistance durability and the ability to deter pathogen inoculum buildup in soil. It is also advisable to use these or any other clubroot-resistant cultivars with some of the cultural-control measures, especially extended crop rotations and possibly soil liming, to reduce inoculum and disease pressure for greater CR performance and durability as well as sustainable clubroot management.

## Figures and Tables

**Figure 1 plants-12-00726-f001:**
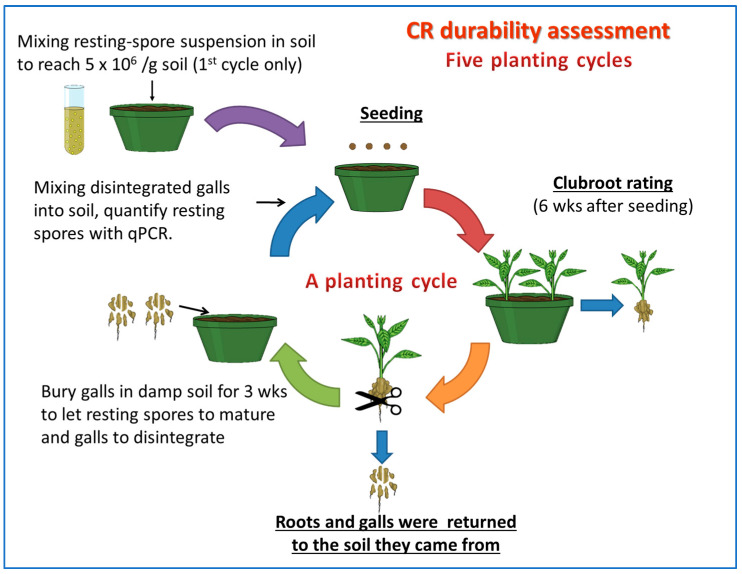
Diagrammatic illustration of the protocol used to conduct five cycles of continuous exposure of selected canola lines carrying different numbers or combinations of CR genes. A resting spore suspension of *Plasmodiophora brassicae* was added to the soil at the beginning of the first cycle to initiate the infection, and canola in subsequent cycles was infected by retained inoculum from previous rounds.

**Figure 2 plants-12-00726-f002:**
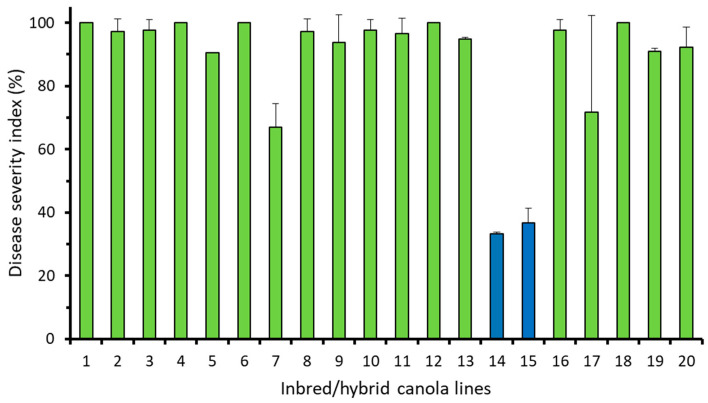
Mean clubroot severity (disease severity index, DSI) in 20 canola lines carrying different numbers or combinations of CR genes in response to L-G3 of pathotype X (data are combined across two repetitions). DSI was substantially lower in lines 14 and 15 (blue bars), relative to others (LSD, *p* < 0.05), and was 100% in the control cultivars Westar (susceptible) and 45H29 (Mendel-based resistance).

**Figure 3 plants-12-00726-f003:**
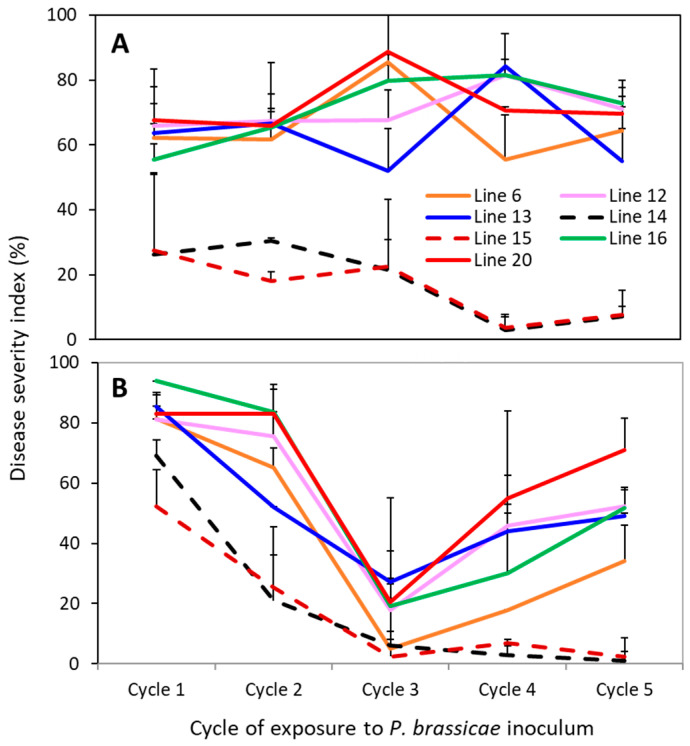
Mean clubroot severity (disease severity index, DSI) on seven selected canola lines carrying different numbers/combinations of CR genes in response to L-G3 of pathotype X in five planting cycles of repetitions 1 (**A**) and 2 (**B**) of the study. The lines 14 (*Rcr1* + *Crr1^rutb^*) and 15 (*CRa^M^* + *Crr1^rutb^*) were moderately resistant (in dash lines), while the rest were susceptible (in solid lines).

**Figure 4 plants-12-00726-f004:**
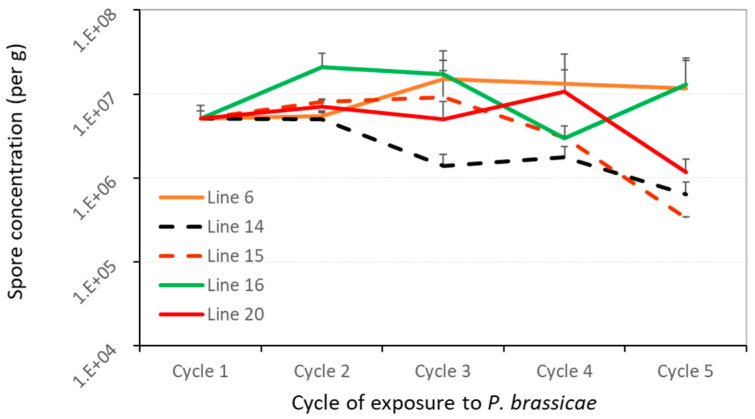
Mean resting spore concentrations in soil quantified with qPCR at the beginning of each cycle of exposure to L-G3 of pathotype X for five selected canola lines carrying a single or double CR genes. Data from two repetitions of the study were combined (*n* = 6). The lines 14 (*Rcr1* + *Crr1^rutb^*) and 15 (*CRa^M^* + *Crr1^rutb^*) were moderately resistant (in dash lines), while the rest were susceptible (in solid lines.

**Table 1 plants-12-00726-t001:** Canola inbred/hybrid lines carrying single, double and triple CR genes tested against three field collections (L-G1, L-G2 and L-G3) of pathotype X of *Plasmodiophora brassicae*.

Canola Lines	CR Genes Involved	# CR Genes	Location of CR Genes
1	*CRa^M^/Crr1^rutb^* ^a^	2	A03/A08
2	*CRa^M^/Crr1^rutb^*	2	A03/A08
3	*CRa^M^/Crr1^rutb^*	2	A03/A08
4	*CRa^M^/Crr1^rutb^*	2	A03/A08
5	*CRa^M^/Crr1^rutb^*	2	A03/A08
6	*CRa^M^/Crr1^rutb^*	2	A03/A08
7	*CRa^M^/Crr1^rutb^*	2	A03/A08
8	*CRa^M^/Crr1^rutb^*	2	A03/A08
9	*Rcr1/CRa^M^*	2	A03/A03
10	*Rcr1//CRa^M^/Crr1^rutb^* ^b^	3	A03//A03/A08
11	*Rcr1//CRa^M^/Crr1^rutb^*	3	A03//A03/A08
12	*Rcr1//CRa^M^/Crr1^rutb^*	3	A03//A03/A08
13	*Rcr1/Rcr1* ^c^	1	A03/A03
14	*Crr1^rutb^/Rcr1*	2	A08/A03
15	*Crr1^rutb^/CRa^M^*	2	A08/A03
16	*CRa^M^*	1	A03
17	*CRa^M^//Rcr1/Rcr1*	2	A03/A03
18	*CRa^M^//CRa^M^/Crr1^rutb^*	2	A03//A03/A08
19	*CRa^M^//Crr1^rutb^/CRa^M^*	2	A03//A03/A08
20	*Crr1^rutb^*	1	A08

^a^ Multiple *CRa^M^/Crr1^rutb^*: Different inbred lines carrying *CRa^M^* or *Crr1^rutb^* used in hybridization. ^b^ Female/male used in crossing, and female//male in second crossing to stack two and three CR genes in hybrids. ^c^ Homozygous in resistance carrying two copies of *Rcr1*.

**Table 2 plants-12-00726-t002:** Canola inbred/hybrid lines carrying different numbers or combinations of CR genes used for studying resistance durability against a field population of *Plasmodiophora brassicae* pathotype X (L-G3) under controlled-environment conditions.

Canola Lines	CR Genes Involved	# CR Genes	Generation	To Pathotype 3 ^a^	Reaction to L-G3 ^b^
6	*CRa^M^/Crr1^rutb^*	2	F_1_	Resistant	Susceptible
12	*Rcr1//CRa^M^/Crr1^rutb^* ^c^	3	F_1_	Resistant	Susceptible
13	*Rcr1/Rcr1*	1	DH ^d^	Resistant	Susceptible
14	*Crr1^rutb^/Rcr1*	2	F_1_	Resistant	Moderately resistant
15	*Crr1^rutb^/CRa^M^*	2	F_1_	Resistant	Moderately resistant
16	*CRa^M^*	1	DH	Resistant	Susceptible
20	*Crr1^rutb^*	1	DH	Resistant	Susceptible

^a^ Pathotype 3 is an old classification based on Williams’ system [9], and has been named as pathotype 3H based on the Canadian Clubroot Differential system [13]. This is the most dominant pathotype found on canola in western Canada. The rating was against the average disease severity index (DSI) on Westar (100%); a ‘resistant’ rating indicated that the average DSI was <30% of that on Westar. ^b^ Based on the resistance assessment shown in Figure 1; a ‘moderately resistant’ rating indicated that the average DSI was between 30% and 50% of that on Westar, and a ‘susceptible’ rating would show >50% DSI. ^c^ Female/male used in crossing, and female//male in second crossing to stack two and three CR genes in hybrids. ^d^ H: Doubled haploid line.

## Data Availability

All datasets from this study are included in the article and Appendix A.

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
