# Peer review of "Canola with Stacked Genes Shows Moderate Resistance and Resilience against a Field Population of Plasmodiophora brassicae (Clubroot) Pathotype X"

_plants, 2023, doi:10.3390/plants12040726_

Round 1

Reviewer 1 Report

MS Title: Canola with stacked genes shows moderate resistance and resilience against a field population of Plasmodiophora brassicae (clubroot) pathotype X. Manuscript ID: Plants-2180380

Clubroot, caused by Plasmodiophora brassicae, is one of the most destructive soil-borne diseases of cruciferous crops throughout the world. The paper aims to evaluate resistance to pathotype X of canola varieties bearing clubroot resistance (CR) genes. The possibility of using pest-resistant varieties is a prerequisite in formulating sustainable cultivation protocols. The strength of the project idea is to evaluate the reliability of the resistance of single and pyramidalized CR genes over time.

Results are relevant showing that under conditions of high disease pressure, the resistance of genotypes carrying two CR genes can mitigate clubroot damage and the build-up of inoculum in the soil.

The research appears original, well-designed and conducted. My opinion is that the results are reproducible based on the details given in MM.

The cited references are appropriate. The cited literature includes a significant number of self-citations. However, they are related to a rich body of the author’s previous research on the topic. I believe that
they are functional to the descriptions along the different sections of the paper.

The weakness that I found in this work does not concern the methodology and the presentation of the results that appear interesting and clear, but rather the crucial point that should be discussed in the discussion and conclusions concerning the potential effect of agronomic measures on the durability of the resistance. Therefore, I recommend the authors interpret the results also in relation to other measures, possibly circular and sustainable that can be integrated with the use of resistant varieties to
pursue disease control while decreasing the risk of breaking resistance, such as the use of amendments or other means capable of increasing soil pH.

Specific comments:
L38: disintegrate, I suggest substituting with “decompose”

L126: explain why the pH and temperature conditions indicated in the experimental protocol were chosen

L279: provide a more detailed hypothesis in the Discussion section regarding the observed trend in cycle 3 as in figure 13. The hypothesis is formulated that the aphid treatment may have influenced the course
of the infection (L405-406). If so, the treatment needs to be described in more detail in MM including the persistence time of the pesticides used.

Author Response

Thanks for the careful review of the manuscript and insightful comments! All suggestions have been addressed in revision and tracked. In the following I just want to respond to two major comments to explain how they were addressed.

  1. For the recommendation to describe the potential effect of agronomic measures on the durability of the resistance, we added a section in Discussion (L327-353) and Conclusion (L372-375) to highlight the importance of using CR varieties with cultural practices, especially a >2-year break from a canola crop.
  2. For the suggestion to provide more information for the interpretation of sudden dip of DSI in the cycle 3 of repeated experiment run, we added the information in Materials and Methods (L177-181) for the circumstance when the insecticides were applied, and additional explanations in Results (L249-252) and Discussion (L356-361) to better interpret the potential impact of insecticide soil drench on clubroot infection. We also cited a paper from Hildebrand and McRae (1998) who reported clubroot suppression by surfactants. 

Author Response

Thanks for your careful review of the manuscript and insightful comments, especially on the pathotype clarification and the question about the sudden dip of DSI in cycle 3 of repeated experiment run. In the following I will just highlight how we have addressed the issues raised:

  1. More information has been added in Introduction (L41-52) to better link old pathotypes (based on Williams') to new/novel pathotype classified using the Canadian Clubroot Differentials, especially for nomenclature changes from pathotype 5, to 5X and finally to X. We have realized that this clarification is important to non-Canadian audience.
  2. The pathotype 3H appeared earlier in Table 2 has been replaced by pathotype 3 to avoid potential confusion. An explanation has been provided in the footnote to clarify the relationship between the two in case some audience might be interested.
  3. L111 (old version): Clarification of pathotype 3 and resistance of CR lines to other pathotypes..... Information has been added in L106-108 (revised version) to provide a bit more details.
  4. Conditions used for growing plants: Additional information has been inserted to provide some reasoning for the conditions used (L125-132).
  5. Question on three collections of pathotype X (L-G1, L-G2 and L-G3) but only the resistance to L-G3 was shown in Figure 2. Response: The results for resistance to L-G1 and L-G2 were shown in Supple. Figure 1 (cited in L225, 227, 232), and the reason to place them in the Supple. information is that the trial for these two collections was conducted only once (as opposed to repeated experiments on L-G3).
  6. Explain why the disease symptoms decreased significantly in cycle 3 of repeated experiment run and then again increased in 4 and 5 (Fig. 3). Response: Additional information has been provided (L177-181, L249-252, L356-361) to provide more background for the scenario, and interpret the results in relation to insecticide soil drench for aphids control. It's more likely the surfactants in the products that suppressed clubroot infection; we have also cited the work of Hildebrand and McRae (1998) in which a soil drench with nonionic surfactants effectively controlled clubroot. We hope these changes have addressed your comments adequately. Thank you again for your time in reviewing this manuscript and your valuable comments/suggestions.